cognition/behaviour

ordinal patterns, multidisciplinary cognition, short-term prediction, natural cognition

**Author for correspondence:**
Yair Neuman
e-mail: yneuman@bgu.ac.il

# Short-term prediction through ordinal patterns

## Yair Neuman[1], Yochai Cohen[2] and Boaz Tamir[3]

[1]Department of Cognitive and Brain Sciences and the Zlotowski Center for Neuroscience, Ben-Gurion University of the Negev, Beer-Sheva 84105, Israel
[2]Gilasio Coding, Tel-Aviv, Israel
[3]The STS Program, Bar-Ilan University, Ramat-Gan, Israel

YN, 0000-0002-3920-0355

Prediction in natural environments is a challenging task, and there is a lack of clarity around how a myopic organism can make short-term predictions given limited data availability and cognitive resources. In this context, we may ask what kind of resources are available to the organism to help it address the challenge of short-term prediction within its own cognitive limits. We point to one potentially important resource: *ordinal patterns*, which are extensively used in physics but not in the study of cognitive processes. We explain the potential importance of ordinal patterns for short-term prediction, and how natural constraints imposed through (i) ordinal pattern types, (ii) their transition probabilities and (iii) their irreversibility signature may support short-term prediction. Having tested these ideas on a massive dataset of Bitcoin prices representing a highly fluctuating environment, we provide preliminary empirical support showing how organisms characterized by bounded rationality may generate short-term predictions by relying on ordinal patterns.

## 1. Introduction

In natural environments, prediction is a cognitive activity that is functionally coupled with a decision-making process, entailing potential risk and gain. For an organism, such an environment is specifically challenging given the environment's non-ergodic nature and the fact that ensemble average and time-average are not the same (e.g. [1–4]). Moreover, and excluding seasonal events, prediction deteriorates as a function of the time distance from the future event, making it difficult to generate long-term predictions. This limit may be attributed to the chaotic dynamic of the observed system and suggests that organisms may have natural adaptive preference for short-term predictions. The existence of an optimal timescale for prediction has surprisingly been exposed in the context of FinTech, where the revolutionary strategy used by the most successful FinTech

innovator—the mathematician Jim Simons—focused on *short-term prediction* [5]. In the context of FinTech, this short-term or *myopic* method of prediction has been supported by a huge amount of data, sophisticated algorithms and massive computational resources. However, in natural environments, there is a lack of clarity around how a myopic organism can generate short-term predictions given noise, limited data availability and limited cognitive resources. If we study short-term prediction through the spectacles of *bounded rationality* [6], we may ask what kind of resources are available to the organism to help it address the challenge of short-term prediction.

In this paper, we try to address this question by pointing to one potentially important resource: *ordinal patterns*, which are extensively used in physics (e.g. [7–9]) but to the best of our knowledge have never been used to model cognitive computational processes in general or short-term prediction in particular. In this paper, we explain the importance of ordinal patterns for short-term prediction and test their use in short-term prediction. Tested on financial data relating to Bitcoin prices, which represent a highly fluctuating environment, the results provide preliminary support showing how organisms with limited cognitive computational resources may naturally and rationally use ordinal patterns for short-term prediction.

# 2. Ordinal patterns and natural constraints

The idea of analysing ordinal patterns may be traced back to the seminal work of Bandt & Pompe [7], where a time series is converted into a series of ordinal permutation patterns. Given a one-dimensional time series $S(t)$ of length $N$, we partition the series into overlapping blocks of length $D$ (the embedding dimension) using a time delay $\tau$. Consider the following time series, which uses $D = 3$ and $\tau = 1$:

$$S(t) = \{34,3,5,23,247,234,12,1,2,3\}.$$

This can be broken down into a sequence of overlapping blocks:

| 34 | 3 | 5 | 23 | 247 | 234 | 12 | 1 | 2 | 3 |
|----|----|----|----|----|----|----|----|----|----|
| 34 | 3 | 5 | 23 | 247 | 234 | 12 | 1 | 2 | 3 |
| 34 | 3 | 5 | 23 | 247 | 234 | 12 | 1 | 2 | 3 |
| 34 | 3 | 5 | 23 | 247 | 234 | 12 | 1 | 2 | 3 |
| 34 | 3 | 5 | 23 | 247 | 234 | 12 | 1 | 2 | 3 |
| 34 | 3 | 5 | 23 | 247 | 234 | 12 | 1 | 2 | 3 |
| 34 | 3 | 5 | 23 | 247 | 234 | 12 | 1 | 2 | 3 |
| 34 | 3 | 5 | 23 | 247 | 234 | 12 | 1 | 2 | 3 |

The elements in each block or vector are then sorted in ascending order and the vector is mapped into one of $D!$ permutations (i.e. $\pi_i$), each representing the ordinal pattern of the elements. For $D = 3$ there are six possible permutations:

$$\pi_1 = \{0,1,2\}$$
$$\pi_2 = \{0,2,1\}$$
$$\pi_3 = \{1,0,2\}$$
$$\pi_4 = \{1,2,0\}$$
$$\pi_5 = \{2,0,1\}$$
$$\pi_6 = \{2,1,0\}.$$

The first partition in the above time series—{34, 3, 5}—is mapped into the permutation pattern $\pi_5 = \{2,0,1\}$; the second partition—{3, 5, 23}—is mapped into $\pi_1 = \{0,1,2\}$; and so on. This results in a *symbolic sequence of permutations*: $\{\pi_s\}s = 1, \ldots, n$. The mapping of the above time series, therefore, produces a time series of permutations:

$$\{34,3,5,23,247,234,12,1,2,3\} \rightarrow \{2,0,1\},\{0,1,2\},\{0,1,2\},\{0,2,1\},\{2,1,0\},\{2,1,0\},\{2,0,1\},\{0,1,2\}.$$

The idea of mapping (i.e. representing) a time series of values into a time series of permutations may be highly relevant to prediction in natural environments, for a very simple reason. Translating a series of

**Table 1.** A list of legitimate transitions from a given permutation type ($D = 3$, $\tau = 1$).

| permutation | legitimate transition to | | |
|---|---|---|---|
| {0,1,2} | {0,1,2} | {0,2,1} | {1,2,0} |
| {0,2,1} | {1,0,2} | {2,0,1} | {2,1,0} |
| {1,0,2} | {0,1,2} | {0,2,1} | {1,2,0} |
| {1,2,0} | {1,0,2} | {2,0,1} | {2,1,0} |
| {2,0,1} | {0,1,2} | {0,2,1} | {1,2,0} |
| {2,1,0} | {1,0,2} | {2,0,1} | {2,1,0} |

numbers into an ordinal pattern shifts the analysis from *absolute values* to *relative rank*. As natural intelligence is sensitive to relations and differences rather than to absolute values [10], this form of representation may naturally support short-term prediction by human and non-human organisms which have limited cognitive resources. As ordinal patterns represent differences in order and are very resilient to noise, they seem to be a natural choice for organisms seeking to represent their environment efficiently. To recall, in natural environments, a difference, rather than an absolute value, may represent an important source of information. For instance, a bacterium sensing a gradient of glucose does so through a *difference of densities* rather than through absolute values—as is demonstrated, for instance, by the Weber–Fechner law in chemotaxis [11].

A highly important aspect of a time series of permutations concerns the *constraints* imposed on the *transition* from permutation $\pi_N$ to the next overlapping permutation: $\pi_{N+1}$. For example, in the above time series of permutations, the first transition is from permutation {2,0,1} to permutation {0,1,2}. While we may naively believe that a transition from each of the six above-mentioned permutation types to any of the six permutation types is possible, this belief is wrong. Each of the six above-mentioned permutation types may move to one of only *three* permutation types. This *inherent* constraint, to be further discussed below, substantially reduces the potential number of transitions from one permutation type to the next, and hence potentially improves the prediction of the $\pi_{s+1}$ permutation in a *symbolic sequence of permutations* $\{\pi_s\}s = 1, \ldots, n$.

As explained by Pessa & Ribeiro [9], constraints imposed on the transition from one permutation to the next do not exist for $D = 2$ but for higher-order embedding dimensions. For $D = 3$ and $\tau = 1$, *which are the focus of our study*, there are only *three legitimate transitions* for each permutation. For example, the second partition that we have previously identified—{3, 5, 23}—is mapped into $\pi_1 = \{0,1,2\}$, where the order of the elements is such that $e_1 < e_2 < e_3$. By definition, the following partition/permutation ($\pi_{N+1}$) overlaps with the previous two elements of the permutation $\pi_N$ and therefore its first two elements *must be ordered* such that $e_1 < e_2$ and the only degree of freedom is left to the third element. Among the six possible permutation types of $D = 3$ and $\tau = 1$, there are only three permutation types consistent with this constraint: {0,1,2}, {0,2,1} and {1,2,0}. What is important to realize is that the constraints imposed on the transition from one permutation to the next significantly reduce the uncertainty associated with the next permutation. This reduction of uncertainty is a highly important cognitive resource (to be discussed further in the next section).

The list of legitimate transitions ($D = 3$, $\tau = 1$) from each permutation type to the next is presented in table 1.

## 3. On the cognitive importance of $D = 3$

At this point, and from the perspective of natural cognitive computation, we may understand the importance of representing a time series using ordinal patterns—specifically those of embedding dimension $D = 3$. Here, we explain the cognitive importance of $D = 3$ (and $\tau = 1$) and why we have used it as the focus of our analysis.

First, $D = 3$ is the first embedding dimension where symmetry-breaking is evident. For $D = 2$, the transitions between permutations {0,1} and {1,0} are unconstrained, meaning that {0,1} → {0,1} or {1,0} and {0,1} → {0,1} or {1,0}. As symmetry-breaking is evident only for $D > 2$, an important source of information may be available for the organism as the number of transitions from one permutation to the next is significantly reduced.

Second, $D = 3$ is the embedding dimension where the number of permutations type to the next is within the 'magic number' (i.e. $7 \pm 2$) identified by [12]. This number exposes a limit of our information-processing capacity and, for $D = 3$, the possible number of permutations falls within this limit.

Third, for $D = 3$, the constraints imposed on the number of transitions are such that the number of possible transitions is reduced *by half*, from six to three. Each permutation can move to only one of three permutation types. For $D = 4$ the number of legitimate transitions is cut only by 0.16, for $D = 5$ it is cut by 0.04 and so on. That is, the maximum relative reduction in uncertainty regarding the transition from one permutation type to the next is evident for transitions between permutations with length 3 (and $\tau = 1$).

Fourth, 'self-loops', meaning transitions from one permutation type to the same permutation type in the next step (e.g. {0,1,2} → {0,1,2}), are allowed only for monotonic increasing and monotonic decreasing permutations (i.e. {0,1,2} and {2,1,0}, respectively). As monotonic increasing and monotonic decreasing sequences/permutations are the simplest instances of upward and downward waves, respectively, we know that when we observe permutations of length 3, the only cases in which the same permutation may appear concatenated are the cases where there are simple upward or downward waves. This knowledge may help us to anticipate upward and downward trends, as further explained below.

In sum, the constraints imposed on transitions from one permutation to another, specifically through the embedding dimension $D = 3$ and $\tau = 1$, may play an important role in supporting short-term prediction of organisms operating with bounded rationality.

# 4. Ordinal patterns and short-term prediction: mathematical, statistical and physical constraints

An organism seeking to perform short-term prediction may observe a time series composed of three successive values only, such as 12, 34 and 265. In this context, one possible challenge is to predict the next (i.e. fourth) value in the time series. A reasonable heuristic for such a short-term prediction is first to turn the time series {12, 34, 265} into an order pattern (i.e. {0,1,2}). In this context, there may be three sources of constraint that may support the organism's prediction of the fourth next value.

The first constraint, and the most important one, is the 'mathematical' constraint on the transition from one permutation type to the next. For the permutations that we discuss in this paper (i.e. $D = 3$, $\tau = 1$), the number of *actual* transitions is only half the number of *potential* transitions, meaning that each permutation can move to only one of three possible permutation types and the uncertainty concerning the next permutation is reduced by half.

For example, given a possible transition from {0,1,2} to one of three possible permutation types ({0,1,2}, {0,2,1} or {1,2,0}), the organism may guess the next possible value in the series. In the case of a transition to {0,1,2}, the next value in the time series (i.e. the fourth value) should be higher than the last value (i.e. 265). However, in the two other cases (involving a transition to {0,2,1} or {1,2,0}), the fourth value in the time series should be lower than the last one. In other words, and from a purely theoretical perspective, when observing a series such as {12, 34, 265}, the next value should involve a *decrease* of the value in two-thirds (i.e. 67%) of the cases. A short-term prediction of the fourth value in a time series, given the preceding three values, may, therefore, be performed by relying on the constraints imposed on the transition from the first permutation.

The mathematical constraint is a potential way to reduce the amount of uncertainty the organism faces. However, here enters the second source of constraint, which we can call 'statistical'. This constraint is introduced by using the above example of observing permutation {0,1,2}. Although, in theory, there is approximately a 67% chance that the fourth value in the time series will be lower than the third value, in practice these chances are determined by the actual *transition probabilities* from one permutation type to the next—in the above example, from {0,1,2} to each of the three possible permutation types. These transition probabilities should ideally be computed on the basis of a long series of events preceding the current time. However, as has counterintuitively been shown in several studies [13,14], the limited capacity of working memory may serve as an amplifier that helps organisms to detect correlations. Given these findings, it is possible that even a small sample of data points preceding a target permutation may be enough to estimate the transition probabilities from one permutation type to the next, in a way that may improve the prediction of the fourth value in the time series.

Learning the transition probabilities from one permutation to the next may provide the organism with another important source of information for short-term prediction. Given that the organism now

'knows' that each permutation can move on to one of only three possible permutation types, if the organism learns the transition probabilities associated with each possible transition, this may further improve its prediction of the fourth value in the series. Moreover, the organism may learn the transition probabilities from a limited time-window *preceding* the time series (i.e. the epoch). Our experiment (described below) shows that the transition probabilities learned from an epoch of only 10 min may be enough to improve short-term prediction.

The third source of information available for short-term prediction is what we may call the 'physical' constraint, as it concerns reversibility. As reversibility is rarely discussed in the context of cognition and natural computation, its relevance to short-term prediction is discussed in the next section.

# 5. Irreversibility, cognition and short-term prediction

We are discussing short-term prediction in the context of organism that is observing a time series of three successive values, translating it into an ordinal pattern and trying to predict the next permutation in order to predict the next (fourth) value in the series. This is not a pointwise prediction but a prediction concerning ordinal relations. In this context, we will discuss irreversibility in a specific sense, but first we address the general meaning of irreversibility.

Irreversibility is evident when, as part of a computation process, the input cannot be fully reconstructed from the output [15]. Some information is lost during this process, which operates under constraints such as memory [16] or abstraction [17]. While irreversibility first appeared in the classical context of thermodynamics, the physics of computation may provide a computational perspective through which we can theoretically create a bridge between the physical realm and the realm of natural cognitive systems. More specifically, the exact pattern of irreversibility that exists in a time series may be indicative of the constraints underlying the observed process, whereas these constraints may be indirectly learned by the organism to help it predict future events. This hypothesis is supported by the observation that 'one should be able to make easier predictions on irreversible processes, where the arrow of time is playing a role, than on reversible ones' [16, p. 1690]. For example, it is suggested by Zanin *et al.* [18, p. 9] that 'irreversibility is a collective (or emergent) phenomenon, which is difficult to see in the dynamics of individual elements, but shows up when considering groups of them'. Following this proposal, the irreversibility score (IRR) of a time series may be indicative of an underlying process of computation operating under potentially learnable constraints such as the transition probabilities between permutation types. More specifically, irreversibility in the context of ordinal patterns may be expressed by the existence of 'symmetric' and 'asymmetric' patterns. Here, we discuss irreversibility in the sense of symmetry and asymmetry, and we explain its relevance to the *specific* context of our study.

Following the seminal work of Elliot [19], we can describe permutation types in terms of upward and downward waves. The typology of upward and downward waves, as proposed by Elliot, perfectly corresponds with the six permutation types of length 3 that are the focus of our interest. In this context, upward waves are expressed by permutation types {0,1,2}, {1,0,2} and {0,2,1}. Downward waves are expressed by permutation types {1,2,0}, {2,1,0} and {2,0,1}.

In this context, irreversibility may be interpreted in terms of asymmetry between the direction of a permutation (e.g. upward) and the direction of its following and overlapping permutation (e.g. downward). This idea can be explained through the permutation type {0,1,2}. For permutation type {0,1,2}, which is an upward wave, a transition is possible to two upward waves/permutations ({0,1,2} and {0,2,1}) and to one downward wave/permutation {1,2,0}. We may consider a transition from an upward wave to another upward wave as symmetric and to a downward wave as asymmetric. We may apply the same idea to a downward wave. In this context, irreversibility may be interpreted in terms of symmetry-breaking and the probability of a transition from one type of wave (e.g. upward) to another type of wave (e.g. downward). For example, when observing permutation {0,1,2}, we may analyse the time series preceding it and ask whether the chance of observing a symmetric transition is higher than that of observing an asymmetric transition. If the chance of observing a symmetric transition is higher, then we can guess that the expected fourth value in the time series is higher than the third value in the time series—or, if the chance of observing an asymmetric transition is higher, the expected fourth value is lower. In this sense, the irreversibility signature of the time series preceding the target permutation may provide us with some information about the approaching (next) permutation.

In this paper, we use two different measures of irreversibility: IRR and IRR_SYM (symmetrical irreversibility score). These measures are detailed in the appendix. In a nutshell, for IRR, we first use

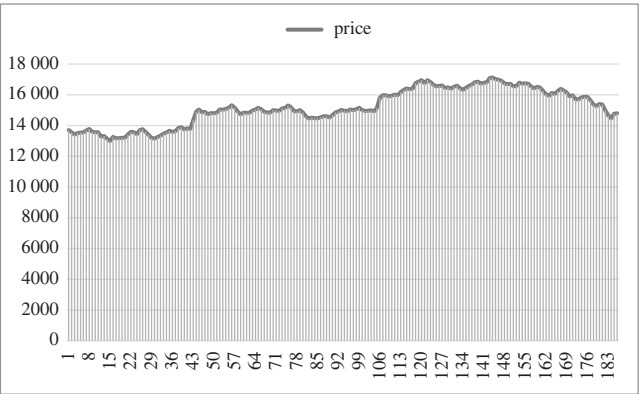

**Figure 1.** Closing prices of Bitcoin averaged per hour (first quarter of January 2018). The horizontal axis represents trading days and the vertical axis the average price.

a time series of data and count the frequencies of each of the ordinal triplets, and next, we read the series backwards and again count the frequencies. We then compare the two distributions of triplets. The distance between the two distributions is a good measure of irreversibility. We compute the distance using Kullback–Leibler divergence. For IRR_SYM, we cluster the six permutation types into two sets, one of an upward type and the other of a downward type. We compute for each triplet the ratio of all its transition probabilities outside its cluster to all its transition probabilities inside its cluster. To allow for smooth reading, these measures are fully detailed in the appendix.

In sum, our main thesis is that an organism with limited cognitive resources can successfully perform short-term prediction by translating (i.e. representing) a time series into overlapping ordinal patterns of length 3 and $\tau = 1$. First, the organism observes a time series of length 3, which can be translated into a permutation. Next, the organism tries to predict the next permutation by relying on three resources: (i) the mathematical constraint limiting the transition of a permutation type to one of only three possible permutation types; (ii) the statistical constraint, where the transition probabilities of permutations in a short time-window preceding the target permutation can be used to predict the next permutation; and (iii) the physical irreversibility constraint, which is expressed in the irrerversibility of the epoch preceding the permutation. Our hypothesis is that when these three sources are used in a machine learning (ML) classifier to model an optimized learning process, we may provide evidence of a successful short-term prediction that relies on ordinal patterns and their transition.

# 6. The experiment

## 6.1. Data

To present, illustrate and test our model, we used a dataset [20] of closing Bitcoin prices per minute for the year 2018 and represented the prices as a single time series. The datafile used in our study is available [21]. It must be emphasized that our interest is fully scientific and we have no practical interest in trading cryptocurrencies. Bitcoin is of specific interest to our paper given its high volatility [22] and accompanying risk [23]. As such, it represents a highly fluctuating environment where decisions on gain and loss can be made.

Figure 1 presents the time series for the first quarter of January 2018, where each data point is the average closing price per hour.

Table 2 presents the relative frequencies of each permutation type in our dataset, as measured using a random sample of 2 765 819 data points.

The transition probability scores (TPSs) of each permutation type are presented in table 3.

## 6.2. Procedure

We applied the following procedure for the data analysis:

1. First, we translated the time series ($S(t)$) into a time series of succeeding permutations ($D = 3$, $\tau = 1$).

**Table 2.** The relative frequencies of each permutation type.

| permutation | % |
|:---:|:---:|
| $\pi_1 = \{0,1,2\}$ | 22 |
| $\pi_2 = \{0,2,1\}$ | 16 |
| $\pi_3 = \{1,0,2\}$ | 15 |
| $\pi_4 = \{1,2,0\}$ | 12 |
| $\pi_5 = \{2,0,1\}$ | 13 |
| $\pi_6 = \{2,1,0\}$ | 22 |

**Table 3.** The transition probabilities of each permutation type. The probabilities appear in curved brackets.

| permutation | transition probability to next permutation | | |
|:---|:---|:---|:---|
| {0,1,2} | {0,1,2} (0.46) | {0,2,1} (0.27) | {1,2,0} (0.27) |
| {0,2,1} | {1,0,2} (0.34) | {2,0,1} (0.19) | {2,1,0} (0.47) |
| {1,0,2} | {0,1,2} (0.38) | {0,2,1} (0.42) | {1,2,0} (0.19) |
| {1,2,0} | {1,0,2} (0.25) | {2,0,1} (0.39) | {2,1,0} (0.36) |
| {2,0,1} | {0,1,2} (0.48) | {0,2,1} (0.27) | {1,2,0} (0.25) |
| {2,1,0} | {1,0,2} (0.32) | {2,0,1} (0.23) | {2,1,0} (0.45) |

Within this time series, we randomly sampled 10 000 cases from each permutation type. Overall, we collected 60 000 observations—i.e. 10 000 observations for each of the six permutation types. These are the *target* permutations.

2. We identified the permutation preceding each one of the target permutations.

3. Next, we identified the time series of length $N$ (i.e. the epoch) *preceding* each target and used these time series for the analysis. There was no overlap between the epochs and our target permutations. We experimented with two different lengths of the epoch: 60 and 10 min.

4. For each epoch, we computed the TPSs for each possible permutation:

$$p\ i{,}j = \frac{\text{number of times permutation } i \text{ is followed by permutation } j}{\text{total number of transitions in the series}}.$$

5. Next, we measured two IRRs of the epoch preceding each target. One irreversibility measure is that proposed by Borges *et al.* [8], here termed IRR; the other is a new measure that we specifically developed for this study (IRR_SYM). These measures and the way they were calculated are described in the appendix.

6. Finally, we used several ML classifiers to classify each target permutation using three main features corresponding to the three above-mentioned constraints: (i) the permutation preceding the target (PRE), (ii) the TPSs learned from the epoch and (iii) the two irreversibility scores (IRR and IRR_SYM).

# 7. Analysis and results

## 7.1. Analysis 1

Given a permutation type (i.e. the target), we identified the preceding permutation overlapping with it (i.e. the PRE) and the epoch preceding it. We used the PRE and the features extracted from the epoch to predict the target. In other words, we tried to successfully classify the permutation type of the target (1–6) using a list of predictors/features. For the first data analysis, we used JASP 0.11.1 (https://jasp-stats.org).

We used 64% of the data to train a ML classifier, 16% of the data for validation and 20% of the cases ($N = 12\,000$) to test the model. The class proportion of each target permutation (1–6) is 17% (rounded). As

in the experiment, we sampled the same proportions of cases from each permutation type; the class proportions of each permutation type in the experiment differ from those that appear in table 3.

Our first analysis used an epoch corresponding to the 60 min preceding our target permutation. We used a boosting regression ML classifier to predict the target permutation based on the above-mentioned features: the permutation preceding our target (PRE), the TPSs that were measured in the epoch, and the two IRRs extracted from the epoch (IRR and IRR_SYM). The ML performance results in terms of averaged precision, recall and area under the curve (AUC) were 40%, 40% and 82%, respectively.

Precision quantifies the number of positive class predictions that actually belong to the positive class—for example, the number of cases in which the ML model correctly classified a permutation as {0,1,2} using the predictive features mentioned above. As precision measures the chance of correctly 'hitting' the target, it may be used as a performance measure to be compared to the baseline. The probability of randomly guessing the permutation type is 0.17. However, the probability of correctly guessing the permutation (i.e. the target) given our predictors (i.e. the features) is 0.40, a result that may be interpreted as a significant improvement in prediction over the base rate.

In terms of the relative influence of each feature, the most important feature was the permutation preceding our target (PRE, i.e. the mathematical constraint), which scored 98.37, followed by the transition probability from permutation type 1 to permutation type 1 as measured in the epoch (influence score = 0.24). The irreversibility measures had relatively lower influence scores (IRR_SYM = 0.07 and IRR = 0.10) but they contributed to the model.

When we repeated the same analysis for an epoch of only 10 min, we gained the same performance results but this time IRR_SYM was the second most important feature (relative influence = 0.19), followed by IRR (relative influence = 0.15). These results are surprising as they show that even a time-window of only 10 min preceding the target permutation allows us to predict it with success. Repeating the same analysis with two additional models (a logistic model and a decision table), this time using the Weka platform (https://www.cs.waikato.ac.nz/ml/weka) (v. 3.8.4), elicited the same results. The fact that the same results were gained for epochs of different sizes and with three different classifiers may support the validity of our results. However, these results may need to be qualified. We modelled the organism using three ML classifiers, which is not a trivial move, as the optimization procedures underlying these classifiers may be beyond the reach of an organism with limited cognitive resources. Therefore, we repeated the analysis for an epoch of 10 min only, this time using a 'lazy' classifier: K-nearest neighbours. Even when using a lazy classifier, the average precision was 24%, which can be interpreted as an improvement in the prediction of 7% over the base rate (17%).

In sum, Analysis 1 shows that given a time series of length 4, identifying the first permutation extracted from the time series and several features characterizing the epoch preceding the target may significantly improve the prediction of the target permutation type.

## 7.2. Analysis 2

Our next analysis examined the ability to predict whether the second permutation in our time series of length 4 is an upward wave following an upward wave or a downward wave following a downward wave. In most of the cases composing our dataset (70%), an upward wave was *not* followed by an upward wave.

We used the same boosting regression model with a binary target variable (1 or 0) indicating whether the target permutation was an upward wave/permutation following an upward wave/permutation. As predictors, we used the PRE and the components composing the IRR_SYM score. The reason for using the components of the IRR_SYM score is that in this context of predicting a trend (e.g. an upward wave following an upward wave), the symmetry-breaking of upward and downward waves may be indicative of a 'trend', a point previously elaborated.

While the probability of observing an upward wave following an upward wave is small ($p = 0.30$), our model gained 65% precision and 80% recall for an epoch of 60 min. To increase the validity of the results, we used another classifier: using a random forest classifier gained 65% precision and 77% recall.

The probability of observing a downward wave following a downward wave was also low ($p = 0.28$), but using the boosting regression model gained 69% precision and 84% recall. Using a random forest classifier gained 69% precision and 80% recall.

Using a random forest classifier for an epoch of 10 min gained 68% precision and 72% recall for the upward wave and 68% precision and 78% recall for the downward wave. Therefore, given an upward or a downward wave, we can significantly improve our prediction of whether the next permutation follows the same trend.

## 7.3. Analysis 3

Next, we tested the ability to predict whether the fourth value in the time series is higher than the third value. We used the binary target UP (1 or 0) to indicate whether the fourth value is higher than the third. As predictors, we used the PRE, the TPS and the irreversibility measures.

The *a priori* probability of observing a fourth value higher than the third in our dataset was $p = 0.50$. Therefore, predicting whether an increase or decrease in value is to be expected in the fourth step of the time series is equivalent to tossing a coin. However, using the boosting regression model, we gained 67% precision and 67% recall, which is a significant improvement in prediction over the baseline set by a random guess. Using a random forest classifier gained 63% precision and 62% recall. Applying the boosting regression model to an epoch of 10 min gained 68% precision and 67% recall.

The practical implications of these results may be illustrated within the context of trading. In terms of Bitcoin value prediction and the simplest strategy (buy low, sell high), knowing that the next value is going to be higher than the current one means that if we buy one unit of the coin at the third step, we should wait for the fourth step in order to sell it and benefit from the 'edge' (i.e. difference in price). Our predictive performance shows that a rational decision can be made based on features extracted from the ordinal patterns.

## 8. Discussion

Our paper examines the ability of a hypothetical organism to perform short-term prediction using limited resources. Adopting the idea of *bounded rationality*, we asked what resources are available to the organism seeking to perform short-term prediction. We proposed a resource that, to the best of our knowledge, has never before been discussed in the context of cognitive research: ordinal patterns. While ordinal patterns might seem to represent a time series in an oversimplified way, this representation is in fact surprisingly powerful and has great explanatory potential in the cognitive sciences. For example, recently it was shown by Olivares *et al.* [24] that ordinal pattern transitions can be used to distinguish between linear and nonlinear dynamical systems even under disturbing noise. The ability to distinguish between different types of dynamical systems may have interesting implications for studying adaptive systems, but in the current study, we have only made a preliminary step in studying the relevance of ordinal patterns to the cognitive sciences.

Our specific proposal is that short-term prediction is possible if a time series is represented as an *order pattern*—specifically, if short segments of the series are converted to permutations of length 3. We have discussed the cognitive computational rationale underlying our suggestions and shown that three limited resources leaning on these ordinal patterns can be used to achieve successful short-term prediction. Our analysis, guided by a cognitive perspective, shows how the appropriate ordinal representation can support short-term prediction, specifically when combined with transition probabilities and the irreversibility signature of even a short time period preceding the prediction. We have also shown that this process illustrates the idea of bounded rationality, as an organism may clearly benefit from short-term prediction of fluctuations in a given dataset. While we have illustrated this rationality in the context of financial data, the same general logic may be applied to many other contexts in which seeking an 'edge' through short-term prediction is of interest. Despite the limits of our study, its importance is in proposing a way to model the short-term prediction of organisms by using ordinal patterns. The study has no pretensions beyond this limited scope.

Data accessibility. The code used to measure the two irreversibility scores of a time series and the data file are available at the Dryad Digital Repository: https://doi.org/10.5061/dryad.vq83bk3r9 [21].

Authors' contributions. Proposed the main thesis: Y.N. Elaborated the thesis, methodology and analysis: Y.N., B.T. Wrote the code for the analysis: Y.C. Wrote the paper: Y.N., B.T. and Y.C.

Competing interests. No competing financial interests.

Funding. This research received no funding.

Acknowledgements. The authors would like to thank the anonymous reviewers for their highly constructive reading of the paper.

# Appendix A

## A.1. The first irreversibility measure

There are various ways of measuring the irreversibility of a time series. Some of them (e.g. [6,8,18]) include the use of Kullback–Leibler divergence (i.e. relative entropy). However, another way, which is specifically appealing in our context, suggests that the irreversibility of a time series may be stochastically measured by comparing the probability of the permutations under the reversal of the time series [8]. For example, the following time series:

$$\{10,4,7,9,10,6,11,3,1\}$$

is translated into the following time series of permutations:

$$\{2,0,1\},\{0,1,2\},\{0,1,2\},\{1,2,0\},\{1,0,2\},\{1,2,0\},\{2,1,0\}.$$

The reversed time series:

$$\{1,3,11,6,10,9,7,4,10\}$$

is then translated into the following time series of permutations:

$$\{0,1,2\},\{0,2,1\},\{2,0,1\},\{0,2,1\},\{2,1,0\},\{2,1,0\},\{1,0,2\}.$$

The probability distributions of the permutations in the original series (i.e. Pd) and the reversed time image (i.e. Pr) appear in table 4.

The IRR can then be computed using Kullback–Leibler divergence (i.e. relative entropy) as follows:

$$D_{KL}(Pd||Pr) = \sum Pd(i) \log \frac{Pd(i)}{Pr(i)},$$

where 0 scores in the vectors are converted to $0 + \delta$. We can then follow this procedure, as proposed by Borges *et al.* [8], to compute the first IRR of a time series. This score is abbreviated as IRR.

## A.2. The second irreversibility measure

The transition from one permutation to another may be informative to the extent that it involves a transition from upward to downward waves. Elliot [19] has identified upward and downward waves that nicely correspond with the permutations (figure 2).

In this figure, the upper leftmost wave corresponds with permutation {1,2,0}, the upper-middle wave corresponds with permutation {2,1,0} and the upper rightmost wave corresponds with permutation {2,0,1}.

We can describe a process of reversibility in terms of a symmetry between a wave and its succeeding pattern. A transition from an upward wave to another upward wave is said to be 'symmetrical', and the same holds for a transition from a downward wave to another downward wave. The possible transitions for downward waves appear in table 5.

The possible transitions for upward waves are shown in table 6.

In this context, irreversibility may be defined as expressing the symmetry-breaking between upward and downward waves, as follows:

1. Given a certain epoch, compute the probability of a transition from an upward wave/permutation to another upward wave/permutation. For example, compute the probability of {0,1,2} moving to {0,1,2} or to {0,2,1}. To do this, we simply count the number of cases in which {0,1,2} moves to {0,1,2} or {0,2,1}, divide the result by the number of all possible transitions and finally multiply the ratio by 100.
2. The probability of a transition from each permutation type (1–6) to another permutation of the same trend (i.e. upward to upward or downward to downward) is indicated as $\pi_i$_SYM (with 'SYM' indicating symmetry).
3. The probability of a transition from each permutation to another permutation of a different trend (e.g. upward to downward) is calculated as $1 - \pi_i$_SYM and indicated as $\pi_i$_ASYM.
4. For permutations $i = 1$ to 6, compute the ratio ($\pi_i$_ASYM/$\pi_i$_SYM) and find the average across the six permutation types.
5. The output irreversibility score of the epoch is indicated as IRR_SYM.

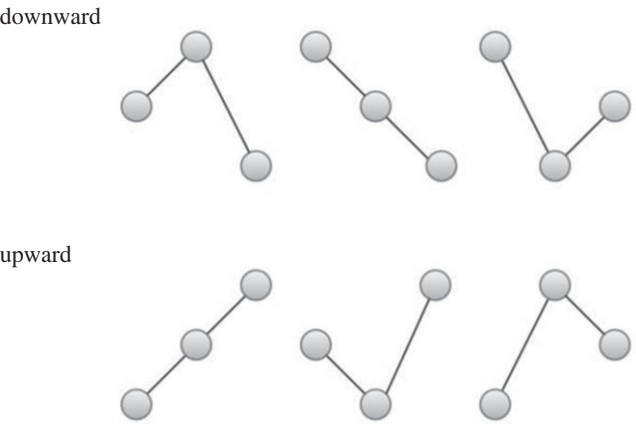

downward

upward

**Figure 2.** Elliot's upward and downward waves.

**Table 4.** The permutations' distributions in the time series and their reverse image.

| $\pi$ | Pd | pr |
|---|---|---|
| 1 | 0.29 | 0.14 |
| 2 | 0 | 0.29 |
| 3 | 0.14 | 0.14 |
| 4 | 0.29 | 0 |
| 5 | 0.14 | 0.14 |
| 6 | 0.14 | 0.29 |

**Table 5.** The possible transitions from a downward wave/permutation to the next downward wave/permutation.

| downward permutation | legitimate transitions to the same type of wave | |
|---|---|---|
| {1,2,0} | {2,0,1} | {2,1,0} |
| {2,1,0} | {2,1,0} | {2,0,1} |
| {2,0,1} | {1,2,0} | — |

**Table 6.** The possible transitions from an upward wave/permutation to the next upward wave/permutation.

| upward permutation | legitimate transition to the same type of wave | |
|---|---|---|
| {0,1,2} | {0,1,2} | {0,2,1} |
| {1,0,2} | {0,1,2} | {0,2,1} |
| {0,2,1} | {1,02} | — |

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
