## [Reviewer comments · Royal Society Open Science]

Review History

RSOS-201011.R0 (Original submission)

Review form: Reviewer 1

Is the manuscript scientifically sound in its present form?

No

Are the interpretations and conclusions justified by the results?

No

Is the language acceptable?

Yes

Do you have any ethical concerns with this paper?

No

Have you any concerns about statistical analyses in this paper?

No

Recommendation?

Major revision is needed (please make suggestions in comments)

Comments to the Author(s)

Review for manuscript RSOS-201011

I have studied the manuscript titled “Learning from Irreversibility” by Yair Neuman, Yochai Cohen and Boaz Tamir. The authors proposed to use the time irreversibility degree in a time series to train a machine learning classifier to forecast herding behaviors. More specifically, the focus is on impulse and corrective waves (increasing and decreasing prices respectively) to characterize the simplest expressions of a collective herding behavior. Inspired on a previous study, they apply the Kullback–Leibler divergence to quantify irreversible structures foregoing to increasing or decreasing values using ordinal patterns. Additionally, they proposed a new metric taking into account transitions between ordinal patterns expecting a more detailed detection of irreversible structures. Application to Bitcoins prices illustrates the performance of this approach.

I believe that the idea to use tools from Information Theory and Physics to get inside of a better understanding of measurements worths it to be addressed, considering the large amount of measurements from numerical and experimental complex systems. Nevertheless, I think the manuscript does not deserve to be published in the present form.

Major considerations are listed below:

- 1.- The authors claimed that PTP is richer and high-dimensional measure of irreversibility. What does “high-dimensional” mean in this context?
- 2.- In the case of having a time series that present weak irreversibility. Which is the significance used to discard spurious irreversible structures?
- 3.- On one hand, the larger the value of D , the more temporal information is considered (in the estimation of the metrics to quantify irreversibility), on the other hand the larger D , the longer increasing(decreasing) patterns can be forecasted. Then, why D is fixed to 3? Does the precision improve with D ?
- 4.- Is finite-size sequences an issue? is the condition $M > (D+1)!$ necessary? What is the minimum past length to obtain a decent forecasting? Or, is there an optimum past length for which the best forecast is obtained?
- 5.- It is well known that finance data could vary their irreversibility, even become locally reversible. Does this feature influence the success of the prediction?
- 6.- Since the forecasting is based on the presence of irreversible structures. Does this approach work better in finance time series coming from emerging countries than developed ones?

Minor considerations:

- 1.- Labels on the horizontal axis in Fig. 1 are unclear.
- 2.- The link provided to get access to the data is not working
- 3.- The authors should consider a recent work that evidences that a metric (PTP) based on permutation transition probabilities (Eq 2) is in fact taking into account non-linear structures [Contrasting chaotic with stochastic dynamics via ordinal transition networks F. Olivares, M. Zanin, L. Zunino, Chaos 30, 063101 (2020)]

Review form: Reviewer 2

Is the manuscript scientifically sound in its present form?

No

Are the interpretations and conclusions justified by the results?

No

Is the language acceptable?

Yes

Do you have any ethical concerns with this paper?

No

Have you any concerns about statistical analyses in this paper?

Yes

Recommendation?

Major revision is needed (please make suggestions in comments)

Comments to the Author(s)

See attached file (Appendix A).

Decision letter (RSOS-201011.R0)

Dear Professor Neuman

The Editors assigned to your paper RSOS-201011 "Learning from Irreversibility" have now received comments from reviewers and would like you to revise the paper in accordance with the reviewer comments and any comments from the Editors. Please note this decision does not guarantee eventual acceptance.

Please submit your revised manuscript and required files (see below) no later than 21 days from today's (ie 02-Nov-2020) date. Note: the ScholarOne system will 'lock' if submission of the revision is attempted 21 or more days after the deadline. If you do not think you will be able to meet this deadline please contact the editorial office immediately.

on behalf of the Associate Editor and Professor Marta Kwiatkowska (Subject Editor)
openscience@royalsociety.org

Associate Editor Comments to Author:

Thank you for your patience while we sought reviewers for your paper - this took longer than we had hoped (no doubt driven in part by the complexities of COVID). Nevertheless, the reviewers consider that your paper requires substantial revisions for it to be considered ready for acceptance. Please carefully revise your paper to take into account their concerns, and ensure that you include a point-by-point response to them, when you resubmit. Good luck and thank you for submitting to Royal Society Open Science!

Reviewer comments to Author:

Reviewer: 1

Comments to the Author(s)

Review for manuscript RSOS-201011

I have studied the manuscript titled "Learning from Irreversibility" by Yair Neuman, Yochai Cohen and Boaz Tamir. The authors proposed to use the time irreversibility degree in a time series to train a machine learning classifier to forecast herding behaviors. More specifically, the focus is on impulse and corrective waves (increasing and decreasing prices respectively) to characterize the simplest expressions of a collective herding behavior. Inspired on a previous study, they apply the Kullback-Leibler divergence to quantify irreversible structures foregoing to increasing or decreasing values using ordinal patterns. Additionally, they proposed a new metric taking into account transitions between ordinal patterns expecting a more detailed detection of irreversible structures. Application to Bitcoins prices illustrates the performance of this approach.

I believe that the idea to use tools from Information Theory and Physics to get inside of a better understanding of measurements worths it to be addressed, considering the large amount of measurements from numerical and experimental complex systems. Nevertheless, I think the manuscript dos not deserve to be published in the present form.

Major considerations are listed below:

- 1.- The authors claimed that PTP is richer and high-dimensional measure of irreversibility. What does "high-dimensional" mean in this context?
- 2.- In the case of having a time series that present weak irreversibility. Which is the significance used to discard spurious irreversible structures?
- 3.- On one hand, the larger the value of D , the more temporal information is considered (in the estimation of the metrics to quantify irreversibility), on the other hand the larger D , the longer

increasing(decreasing) patterns can be forecasted. Then, why D is fixed to 3? Does the precision improve with D?

4.- Is finite-size sequences an issue? is the condition $M > (D+1)!$ necessary? What is the minimum past length to obtain a decent forecasting? Or, is there an optimum past length for which the best forecast is obtained?

5.- It is well known that finance data could vary their irreversibility, even become locally reversible. Does this feature influence the success of the prediction?

6.- Since the forecasting is based on the presence of irreversible structures. Does this approach work better in finance time series coming from emerging countries than developed ones?

Minor considerations:

1.- Labels on the horizontal axis in Fig. 1 are unclear.

2.- The link provided to get access to the data is not working

3.- The authors should consider a recent work that evidences that a metric (PTP) based on permutation transition probabilities (Eq 2) is in fact taking into account non-linear structures [Contrasting chaotic with stochastic dynamics via ordinal transition networks F. Olivares, M. Zanin, L. Zunino, Chaos 30, 063101 (2020)]

Reviewer: 2

Comments to the Author(s)

See attached file.

===PREPARING YOUR MANUSCRIPT===

===PREPARING YOUR REVISION IN SCHOLARONE===

Author's Response to Decision Letter for (RSOS-201011.R0)

See Appendix B.

RSOS-201011.R1 (Revision)

Review form: Reviewer 1

Is the manuscript scientifically sound in its present form?

Yes

Are the interpretations and conclusions justified by the results?

No

Is the language acceptable?

Yes

Do you have any ethical concerns with this paper?

No

Have you any concerns about statistical analyses in this paper?

Yes

Recommendation?

Accept with minor revision (please list in comments)

Comments to the Author(s)

Review for manuscript RSOS-201011-R1

I have studied the new version of the manuscript retitled "Short-Term Prediction through Ordinal Patterns" by Yair Neuman, Yochai Cohen and Boaz Tamir. It is clear that the authors have learned from the criticisms of the reviewers and they have submitted an almost new study. Even though the reasons listed for the selection of $D=3$ are rather heuristic I believe they are valid as a justification. Additionally, fixing $D=3$ discard finite-size issues.

Albeit, the overlapping procedure while observing transitions between permutations bias the estimation of its probabilities of occurrence, it seems to "improve" the prediction of the next permutations. Nevertheless, It is known that when overlapping is considered, the transitions probabilities between ordinal patterns from a totally uncorrelated data are not equiprobable ($1/(D!*D!)$) as expected or wanted. Said that, is your methodology able to short-term predict permutations in a uncorrelated data? I believe this is important to clarify. Then I wonder, if you can predict permutation in a white noise maybe you are observing spurious temporal correlations due to the overlapping.

Related to the previous point. I'm not a trader but I'm very familiarized with Bitcoin prices. I believe that observing its price at very high frequencies (less than one hour between prices) gives a random dynamics. So, are you predicting permutations in a random price time series? Labels of Fig. 1 are still unclear since they overlap as you observe rightwards. The sentence "The figure illustrate the complexity of the signal that we analyzed" is unclear, since the Fig. 1 depicts just the price evolution over a time window.

Review form: Reviewer 2

Is the manuscript scientifically sound in its present form?

Yes

Are the interpretations and conclusions justified by the results?

Yes

Is the language acceptable?

Yes

Do you have any ethical concerns with this paper?

No

Have you any concerns about statistical analyses in this paper?

No

Recommendation?

Accept as is

Comments to the Author(s)

I thank the authors for taking my comments seriously. The current version of the paper addresses my concerns and I recommend the paper for publication.

Decision letter (RSOS-201011.R1)

This year has been very difficult for everyone, and we want to take the opportunity to thank you for your continued support in 2020.

The Royal Society Open Science editorial office will be closed from the evening of Friday 18 December 2020 until Monday 4 January 2021. We will not be responding during this time. If you have received a deadline within this time period, please contact us as soon as possible to allow us to extend the deadline. If you receive any automated messages during this time asking you to meet a deadline, we offer apologies and invite you to respond after the festive period or during normal working hours.

With our best for a peaceful festive period and New Year, and we look forward to working with you in 2021.

Dear Professor Neuman

On behalf of the Editors, we are pleased to inform you that your Manuscript RSOS-201011.R1 "Short-Term Prediction through Ordinal Patterns" has been accepted for publication in Royal Society Open Science subject to minor revision in accordance with the referees' reports. Please find the referees' comments along with any feedback from the Editors below my signature.

Please submit your revised manuscript and required files (see below) no later than 7 days from today's (ie 21-Dec-2020) date. Note: the ScholarOne system will 'lock' if submission of the revision is attempted 7 or more days after the deadline. If you do not think you will be able to meet this deadline please contact the editorial office immediately.

on behalf of Mr Andrew Dunn (Associate Editor) and Marta Kwiatkowska (Subject Editor)
openscience@royalsociety.org

Associate Editor Comments to Author (Mr Andrew Dunn):
Associate Editor: 1
Comments to the Author:
(There are no comments.)

Associate Editor: 2
Comments to the Author:
(There are no comments.)

Reviewer comments to Author:
Reviewer: 1

Comments to the Author(s)
Review for manuscript RSOS-201011-R1

I have studied the new version of the manuscript retitled "Short-Term Prediction through Ordinal Patterns" by Yair Neuman, Yochai Cohen and Boaz Tamir. It is clear that the authors have learned from the criticisms of the reviewers and they have submitted an almost new study. Even though the reasons listed for the selection of $D=3$ are rather heuristic I believe they are valid as a justification. Additionally, fixing $D=3$ discard finite-size issues.

Albeit, the overlapping procedure while observing transitions between permutations bias the estimation of its probabilities of occurrence, it seems to “improve” the prediction of the next permutations. Nevertheless, It is known that when overlapping is considered, the transitions probabilities between ordinal patterns from a totally uncorrelated data are not equiprobable ($1/(D!*D!)$) as expected or wanted. Said that, is your methodology able to short-term predict permutations in a uncorrelated data? I believe this is important to clarify. Then I wonder, if you can predict permutation in a white noise maybe you are observing spurious temporal correlations due to the overlapping.

Related to the previous point. I’m not a trader but I’m very familiarized with Bitcoin prices. I believe that observing its price at very high frequencies (less than one hour between prices) gives a random dynamics. So, are you predicting permutations in a random price time series?

Labels of Fig. 1 are still unclear since they overlap as you observe rightwards.

The sentence “The figure illustrate the complexity of the signal that we analyzed” is unclear, since the Fig. 1 depicts just the price evolution over a time window.

Reviewer: 2

Comments to the Author(s)

I thank the authors for taking my comments seriously. The current version of the paper addresses my concerns and I recommend the paper for publication.

===PREPARING YOUR MANUSCRIPT===

===PREPARING YOUR REVISION IN SCHOLARONE===

Author's Response to Decision Letter for (RSOS-201011.R1)

See Appendix C.

Decision letter (RSOS-201011.R2)

Dear Professor Neuman,

It is a pleasure to accept your manuscript entitled "Short-Term Prediction through Ordinal Patterns" in its current form for publication in Royal Society Open Science.

on behalf of the Associate Editor, and Professor Marta Kwiatkowska (Subject Editor)
openscience@royalsociety.org

Appendix A

Summary

The paper draws upon the idea of irreversibility in a time-series in order to build a time-series prediction model that looks at patterns of permutations in data. They identify the parameters of a Markov chain that describes the transitions between permutations. They hypothesize that monotone increasing sequences (MIS) and monotone decreasing sequences (MDS) should be easier to predict based on models of herding behavior. They use this to build a random forest and logistic regression model that learns to predict the probability of a MIS from the identified transition matrix is able to achieve reasonable levels of precision.

General Comments

At its core, the paper has identified an interesting pattern in financial time series data. The basic ideas of reducing the time-series to an encoding space of permutations seems like a very useful type of feature. The results have convinced me there's something interesting in the approach presented. However, the paper interleaves this idea with handwavy arguments about connections to intelligence and cognitive systems that are repeated often but poorly supported. This has the effect of making it quite hard to understand the approach that was actually followed.

This is not to say that the author's positions are wrong, but the reasoning behind the authors claims and the connections to supporting evidences are poorly described in the submitted manuscript. I think my point is illustrated by my proposed change to 1.10-1.17 on page 18:

Original: "We have shown that the new order parameter characterized by the rich signature of irreversibility can be used by an intelligent agent, as modelled through the machine learning classifier, to gain a better forecast of future events -- specifically, the forecasting of an approaching MIS of length 3."

Proposed: "We have shown that the new order parameter characterized by permutation transition probabilities can be used by a random forest classifier to gain a better forecast of an approaching monotone increasing sequence in financial time-series data. This has implications for theories of intelligence because... "

The authors have found an interesting pattern. It is true that it appears to support (or may be base on insights from) Elliot Wave Theory, but the authors should clearly separate the aspects of the paper:

- 1) The features used as inputs for the ML classifier, the classification method used, and the evaluation strategy. The current description is spaced throughout the paper and important details are missing (e.g., the definition of IRR appears to be missing from line bottom of page 8). I also think the ML evaluation here is weak (e.g., this would be far below the bar for NeurIPS or

ICML in terms of evaluation approach and description, I provide some suggestions below).

2) The motivation for the model presented. Its connections to physics and an overview of irreversibility. The current paper attempts to do this, but seems to rely on a fair amount of familiarity with financial time-series. If the intent is for this to be accessible to researchers who work more generally on A.I. or cognitive science, then background needs to be largely rewritten to target a broader audience. (If my assumptions about the audience are wrong, please disregard this comment.)

3) The connections to intelligent agents and the relationship between these results and the authors hypothesis that 'the exact pattern of irreversibility that exists in a time-series may be indicative of the constraints underlying the observed process and these constraints may be indirectly learned by an intelligent agent.' In my reading of the paper, this claim is grossly undersupported. I recommend weakening it substantially and moving it to the discussion. The paper does not show this result in its current form. The experiments show that learning a Markov transition matrix allows you to predict the next state better than random.

Overall, I think that addressing these concerns requires a major revision to the paper to clearly identify the experiment that was run and the connections it has to properties of cognition and intelligence. In its current form, the paper oscillates between these three points and it makes it hard to understand and evaluate the result.

Specific Comments

Abstract

The abstract assumes a large level of familiarity with the ideas and concepts used in the paper. It references hypothesis 1 and hypothesis 2 without defining them (and so can not be evaluated without actually reading the paper). Some specific questions the authors should address:

- how does the analysis lead to the conclusions claimed
- how does IRR relate to "basic physical concepts" and "foundations of cognitive intelligent systems"

Please add more details, define or avoid jargon, and fix clarity issues.

Introduction

I found the first paragraph fairly unhelpful at introducing the basic ideas. As a researcher in A.I., I had to spend a fair amount of time

doing background research on irreversibility to get the background needed. The concept isn't that complex and I believe the authors can/should provide an accessible overview.

pg. 2:

1.39: claim is unsupported or unclear

1.41-42: 'irreversibility may be indicative of... computation' is vague and appears tautological

pg. 3:

1.10: "such as that evident" awkward

1.15-1.40: the example is hard to follow unless you already understand the underlying concepts (in which case it is not needed in this detail)

1.40-45: clarity issues; please more clearly explain how these are examples of herding.

pg. 4:

1.3: claim needs support. appear to rely on the claim that predicting financial time-series is evidence of intelligence. This is the primary connection the authors overly strong claims about implications for the study of intelligence.

1.21-31: The terms in this hypothesis need to be defined and the hypothesis need to be made much more specific. Before stating the hypothesis please provide more precise descriptions of how IRR is measured and what is meant by non-linear patterns of irreversibility. Consider including a comparison with what a linear pattern of irreversibility would look like.

Section 2

pg 5.

1.21-26: awkward/unclear wording

1.57: example appears to be missing from pdf

pg. 6 I found this example confusing, but that may have been because the matrix of data was missing.

Section 3

1.56: "Some of them ... divergence" -- awkward and not informative

pg. 7

Consider moving this example (which I found quite helpful) earlier in the paper.

1.59: The definition is cut off. Either it's a generic definition of KL divergence (in which case it is unhelpful) or it precisely defines the

IRR statistic used (which is crucial to include).

pg. 8

I found this to be an incredibly jargony description of a very simple idea. Consider grounding this specifically in the math of Markov chains (even if a non-linear* pattern is what will be eventually used). Ultimately, this is just suggesting that the transition distribution of the Markov chain between the discrete permutations is useful.

(NB: I'm not actually sure what the author mean by 'non-linear' signature in this context. This is also something that should be clarified).

pg. 9

H2 should be much more specific. In particular, 'machine learning classifier modelling an intelligent agent' should be changed to 'machine learning classifier' at a minimum.

Section 4

This is not easily reproducible based on the paper alone. The source code is, of course helpful, but the paper should be able to describe the approach clearly enough that the experiments can be replicated in the reader's modelling language of choice. That is difficult in the current manuscript.

pg. 12

1.27-29: the experiments do not 'address all concerns about overfitting.' This is an overly strong claim worded awkwardly.

pg. 13

1.19-22: clarity issues, provide a more detailed description of your evaluation approach.

pg. 16

1.55: Please explain why you switched to logistic regression. Why not use both models for all experiments?

pg. 17

1.59: Why not report results for more than one epoch size. The chosen result seems arbitrary. At a minimum, it seems possible to do ~10 different sizes.

11/13/2020

Dear Professor Marta Kwiatkowska (Subject Editor)

Royal Society Open

Re. RSOS-201011 ‘Learning from Irreversibility’

We would like to thank the two reviewers for their critical and highly constructive review of the paper. Their reviews have led not just to a substantial revision of the paper but to a *total re-writing*, from the paper’s title to the data analysis.

What did we understand from the reviews? We understood that our paper lacked a clear theoretical skeleton through which the analysis and results could be interpreted. Therefore, in the re-written version, we first locate our paper within the context of interdisciplinary cognition. This is not a paper on artificial intelligence (AI), machine learning (ML) or physics. Our paper actually focuses on ‘short-term prediction through ordinal patterns’, which is the new title of the paper. In this context, we present the following arguments:

1. We explain that prediction in natural environments is a challenging task.
2. We show that there is a lack of clarity around how a myopic organism can make short-term predictions.
3. We point to one potentially important source of information/constraint, which is *ordinal patterns*.

4. We explain the importance of ordinal patterns in short-term prediction and show how natural constraint on short-term prediction are imposed through:
 - ordinal patterns;
 - their transition probabilities; and
 - their ‘irreversibility signature’.
5. Having tested these ideas on a massive dataset of Bitcoin prices representing a highly fluctuating environment, we provide preliminary empirical support showing how organisms characterised by bounded rationality may generate short-term predictions by relying on ordinal patterns.
6. In this context, we explain the meaning of irreversibility by diverging from the physical perspective used in the previous version of the paper, and by discussing irreversibility and the way it may support short-term prediction through ordinal patterns.
7. In addition, we introduce a new measure of irreversibility specifically designed for use in the context of our study.

Given this new reframing, we clearly present the procedure:

1. First, we translated a time-series of the Bitcoin prices for 2018 into a time-series of succeeding permutations ($D = 3, \tau = 1$).
2. From this time-series, we randomly sampled 10,000 cases from each permutation type. Overall, we collected 60,000 observations – i.e. 10,000 observations for each of the six permutation types.
3. We identified the permutation preceding each appearance of one of the six permutation types.

4. Next, and for each appearance of each permutation type (i.e. 1–6), we identified the time-series of length N (i.e. the Epoch) preceding the permutation and used it for the analysis.
5. We used two Epochs: one of length 60 minutes and the other of length 10 minutes.
6. For each EPOCH, we computed the transition probability scores (TPSs) for each possible permutation.
7. Next, we measured two irreversibility scores of the Epoch preceding the each appearance of one of the six permutation types. One irreversibility measure is the one we used in the previous version of the paper and (i.e. IRR) and the other one is a new measure that we specifically developed for this study (IRR_SYM).
8. Finally, we used several ML classifiers to model the prediction process, specifically by classifying the permutation type (1-6) using three main features: (1) the permutation preceding our target permutation (PRE), (2) the TPSs learned from the Epoch and (3) the two irreversibility scores (IRR and IRR_SYM).

Using this procedure, our analysis shows how short-term prediction is possible through the use of ordinal patterns and their different measurements. For example, in Analysis 2, we examined the ability to predict whether an upward wave of prices follows an upward wave of prices. The probability of observing an upward wave of prices following an upward wave of prices was very small ($p = 0.30$), but our ML model gained 65% precision and 80% recall in predicting an upward wave using the components of the new proposed irreversibility measure.

Given this re-writing of the paper, the overwhelming majority of the reviewers' concerns seem to be addressed. For example:

Reviewer 1 asks: Why D is fixed to 3?

Response: We wrote a whole new section explaining the specific importance of $D = 3$:

“At this point, and from the perspective of natural cognitive computation, we may understand the importance of representing a time-series using ordinal patterns – specifically those of embedding dimension $D = 3$. Here we explain the cognitive importance of $D = 3$ (and $\tau = 1$) and why we have used it as the focus of our analysis.

First, $D = 3$ is the first embedding dimension where symmetry-breaking is evident. For $D = 2$, the transitions between permutations $\{0,1\}$ and $\{1,0\}$ are unconstrained, meaning that $\{0,1\} \rightarrow \{0,1\}$ or $\{1,0\}$ and $\{0,1\} \rightarrow \{0,1\}$ or $\{1,0\}$. As symmetry-breaking is evident only for $D > 2$, an important source of information may be available for the organism as the number of transitions from one permutation to the next is significantly reduced.

Second, $D = 3$ is the embedding dimension where the number of possible transitions from one permutation type to the next is within the ‘magic number’ (i.e. 7 plus or minus 2) identified by [12]. This number exposes a limit of our information-processing capacity and, for $D = 3$, the possible number of transitions from one permutation to the next falls within this limit. Therefore, the potential number of transitions from one permutation type to the next seems to be natural when using $D = 3$.

Third, for $D = 3$, the constraints imposed on the number of transitions are such that the number of possible transitions is reduced *by half*, from six to three. Each

permutation can move to only one of three permutation types. For $D = 4$ the number of legitimate transitions is cut only by 0.16, for $D = 5$ it is cut by 0.04 and so on. That is, the maximum relative reduction in uncertainty regarding the transition from one permutation type to the next is evident for transitions between permutations with length 3 (and $\tau = 1$).

Fourth, ‘self-loops’, meaning transitions from one permutation type to the same permutation type in the next step (e.g. $\{0,1,2\} \rightarrow \{0,1,2\}$), are allowed only for monotonic increasing and monotonic decreasing permutations (i.e. $\{0,1,2\}$ and $\{2,1,0\}$ respectively). As monotonic increasing and monotonic decreasing sequences/permutations are the simplest instances of upward and downward waves, respectively, we know that when we observe permutations of length 3, the only cases in which the same permutation may appear concatenated are the cases where there are simple upward or downward waves. This knowledge may help us to anticipate upward and downward trends, as further explained below.

In sum, the constraints imposed on transitions from one permutation to another, specifically through the embedding dimension $D = 3$ and $\tau = 1$, may play an important role in supporting short-term prediction of organisms operating with bounded rationality”.

This reviewer also asks: Is finite-size sequences an issue?

Response: We explain that our proposal and analysis are specifically relevant to short-term prediction using finite-size and short sequences. We back up our arguments with findings from the cognition and cognitive sciences.

Reviewer 1: It is well known that finance data could vary their irreversibility, even become locally reversible. Does this feature influence the success of the prediction?

Response: Not at all. We suggest that irreversibility can be interpreted in terms of breaking the symmetry of transition between permutations and explain how irreversible and reversible patterns may support short-term prediction:

“Following the seminal work of Elliot [19], we can describe permutation types in terms of upward and downward waves. The typology of upward and downward waves, as proposed by Elliot, perfectly corresponds with the six permutation types of length 3 that are the focus of our interest. In this context, upward waves are expressed by permutation types $\{0,1,2\}$, $\{1,0,2\}$ and $\{0,2,1\}$. Downward waves are expressed by permutation types $\{1,2,0\}$, $\{2,1,0\}$ and $\{2,0,1\}$.

In this context, irreversibility may be interpreted in terms of asymmetry between the direction of a permutation (e.g. upward) and the direction of its following and overlapping permutation (e.g. downward). This idea can be explained through permutation type $\{0,1,2\}$. For permutation type $\{0,1,2\}$, which is an upward wave, a transition is possible to two upward waves/permutations ($\{0,1,2\}$ and $\{0,2,1\}$) and to one downward wave/permutation $\{1,2,0\}$. We may consider a transition from an upward wave to another upward wave as symmetric and to a downward wave as asymmetric. We may apply the same idea to a downward wave. In this context, irreversibility may be interpreted in terms of symmetry-breaking and the probability of a transition from one type of wave (e.g. upward) to another type of wave (e.g. downward). For example, when observing permutation $\{0,1,2\}$, we may analyse the time-series preceding it and ask whether the chance of observing a symmetric transition

is higher than that of observing an asymmetric transition. If the chance of observing a symmetric transition is higher, then we can guess that the expected fourth value in the time-series is higher than the second value in the time-series – or, if the chance of observing an asymmetric transition is higher, the expected fourth value is lower. In this sense, the irreversibility signature of the time-series preceding the target permutation may provide us with some information about the approaching (next) permutation”.

Reviewer: Since the forecasting is based on the presence of irreversible structures. Does this approach work better in finance time-series coming from emerging countries than developed ones?

Response: We cannot answer this question as we have no specific knowledge or interest in the cross-cultural variability of financial data. We explain the reason for using this data:

“Bitcoin is of specific interest to our paper given its high volatility [22] and accompanying risk [23]. As such, it represents a highly fluctuating environment where decisions on gain and loss can be made”.

Minor considerations of reviewer 1:

1. Labels on the horizontal axis in Fig. 1 are unclear.

Revision: Corrected.

2. The link provided to get access to the data is not working.

Revision: Checked.

3. The authors should consider a recent work that evidences that a metric (PTP) based on permutation transition probabilities (Eq 2) is in fact taking into account non-linear structures [Contrasting chaotic with stochastic dynamics via ordinal transition networks F. Olivares, M. Zanin, L. Zunino, Chaos 30, 063101 (2020)].

Revision: Read and cited this interesting paper.

Reviewer 2:

The main concern of the reviewer was lack of clarity, ‘making it quite hard to understand the approach that was actually followed’.

Revision: We took the reviewer’s general comments extremely seriously, and this led to us re-writing the paper, as explained above. We hope that this re-written version now presents a coherent and well-supported argument. To make it easy to understand the ‘approach’, we have focused emphasized the cognitive science perspective, enhanced the prominence of irreversibility in our presentation and data analysis; and moved the operational definition of irreversibility to the appendix in order to avoid cognitive overload and facilitate a smooth reading of the text.

Regarding the reviewer’s comments concerning AI and ML: this is not a paper in ML or AI. The first author is a cognitive scientist with extensive background in using ML and AI, the second author is a quantum physicist and the third is a software engineer with a math degree. None of us identifies himself as an AI or ML person, and we hope that the new version of the paper corrects this misleading impression.

The reviewer commented that ‘the experiments show that learning a Markov transition matrix allows you to predict the next state better than chance’. This focus motivated us to re-analyze the data and to present them accordingly, and we hope that our focus and analysis are now clear. We have also analyzed epoch sizes of 60 and 10 minutes to address this final comment.

In sum, we have totally re-written the paper and hope that this revision addresses all of the reviewers’ concerns and that the paper is publishable.

Sincerely yours,

Prof. Yair Neuman

Dear Professor Marta Kwiatkowska (Subject Editor)

Royal Society Open

Re. RSOS-201011 " Short-Term Prediction through Ordinal Patterns"

We were pleased to hear that the paper has been accepted with “minor revision” and would like to thank the reviewers and the editors for the work invested in our paper.

- Associate Editors 1 and 2 have no comments.
- Reviewer 2 thanks us for taking his comments seriously and recommend the paper for publication.
- Reviewer: 1 still have a few comments:

Comment 1: “The **overlapping procedure** while observing transitions between permutations **bias** the estimation of its probabilities of occurrence, it seems to “**improve**” the prediction of the next permutations” (our emphasis).

Our explanation: We agree with the reviewer that the overlapping permutations improves prediction. However, this is not a source of bias but one of our main arguments that representing a time-series of length 4 through overlapping permutations improves the prediction of the fourth value. This is not a bias but the mathematical constraint (see section 4) the organism may use for prediction. In other words, the idea of representing a time-series of overlapping permutations is at the *core* of our *cognitive-oriented theorization*.

Comments 2 & 3:

“It is known that when overlapping is considered, the transitions probabilities between ordinal patterns from a totally **uncorrelated** data are not equiprobable ($1/(D!*D!)$) as *expected* or *wanted*” (Our emphasis).

“I wonder, if you can predict permutation in a white noise maybe you are observing spurious temporal correlations due to the overlapping. Related to the previous point. I’m not a trader but I’m very familiarized with Bitcoin prices. I believe that observing its price at very high frequencies (less than one hour between prices) gives a **random dynamics**. So, are you predicting permutations in a **random price time series**?” (our emphasis).

Our explanation: The first question is whether our data involves random uncorrelated data. There are several pieces of evidence showing this is not the case. Let us start with a new analysis we have made to address this concern. First, we inspected the spectrum of the bitcoin data. We took the set of bitcoin prices for January 2018. We computed a discrete fast Fourier transform of the data. Without going into too much details, it turns out that the spectrum has very high frequencies and very low frequencies, and almost zero power at moderate frequencies. Clearly *this assures that the data is not a white noise*. See the following figures:

Fig 1. Low frequency spectrum

Fig 2. High frequency spectrum

We don't include this analysis in the final version as it aims to address the reviewer's concern only.

Second. It is important to clarify that our data set is the time-series of overlapping *permutations*. We use the permutations for the ML models. In this context, our data set is not random as proven by the transition probabilities between permutations and the permutations' different frequencies (see Tables 2 and 3). Therefore, the context is not the one of a random time-series of raw data points or permutations, where data points are uncorrelated.

As hopefully clarified before, the reviewer's concern is addressed as our data set doesn't involve random data points. Regarding the "prediction from noise" concern. We apply prediction to a very short time-series where the notion of randomness is irrelevant given the transition probabilities between permutations. As our time series of permutations is not random, the answer to the question: "are you predicting permutations in a random price time series?" is negative.

Finally, as the overlapping permutations (i.e. the mathematical constraint we explain in the paper) limits the potential space of transitions in an ordered way and as the empirically observed transition probabilities are not random, the concern of randomness and prediction is fully addressed. It is common in some context to compare the performance to a surrogate random time series. See my paper:

Neuman, Y., Marwan, N., & Cohen, Y. (2014). Change in the Embedding Dimension as an Indicator of an Approaching Transition. *PloS one*, 9(6), e101014.

However, this comparison is irrelevant to the context of the current paper as explained above.

Concern 4:

Labels of Fig. 1 are still unclear since they overlap as you observe rightwards.

The sentence "The figure illustrates the complexity of the signal that we analyzed" is unclear, since the Fig. 1 depicts just the price evolution over a time window.

Our revision: The problematic sentence has been deleted.

The labels of Figure 1 are very simple: “The horizontal axis represents trading days” (from 1 to 183) and “the vertical axis the average price” but we will be happy to get any suggestion how to revise them in a friendlier way.

In sum, we thank the reviewer for raising some interesting concerns and hope that the above explanations address them all. We hope that the required “minor revision” is now satisfying, that the concerns of Reviewer 2 have been addressed, and that the paper is finally publishable.

Sincerely yours,

Prof. Yair Neuman